# Wear Behavior of Graphene-Reinforced Alumina–Silicon Carbide Whisker Nanocomposite

**DOI:** 10.3390/nano9020151

**Published:** 2019-01-25

**Authors:** Anton Smirnov, Pavel Peretyagin, Nestor Washington Solís Pinargote, Iosif Gershman, Jose F. Bartolomé

**Affiliations:** 1Moscow State University of Technology “STANKIN”, Vadkovsky per. 1, 127055 Moscow, Russia; p.peretyagin@stankin.ru (P.P.); nw.solis@stankin.ru (N.W.S.P.); isgershman@gmail.com (I.G.); 2Instituto de Ciencia de Materiales de Madrid (ICMM), Consejo Superior de Investigaciones Científicas (CSIC), C/ Sor Juana Inés de la Cruz 3, 28049 Madrid, Spain

**Keywords:** wear-resistance, friction coefficient, ceramic nanocomposites, graphene

## Abstract

In the present work, the tribological properties of graphene-reinforced Al_2_O_3_-SiCw ceramic nanocomposites fabricated by spark plasma sintering were studied against alumina ball. Compared with pure ceramic, the wear resistance of these nanocomposites was approximately two times higher regardless of the applied load. It was confirmed by Raman spectroscopy that the main factor for the improvement of the wear resistance of the Al_2_O_3_-SiCw/Graphene materials was related to the formation of protecting tribolayer on worn surfaces, which leads to enough lubrication to reduce both the friction coefficient, and wear rate.

## 1. Introduction

Alumina ceramics with their high hardness, chemical inertness, high wear, and corrosion resistance play an important role in different areas. Nevertheless, moderate strength, toughness, and flaw-tolerance of alumina limit their applications. Thereby, the presence of any type of bulk discontinuities or tiny defects may reduce their reliability [1,2]. In order to overcome this limitation, several options have been proposed, for example, the incorporation of SiC whiskers. The primary toughening mechanisms of SiC whisker reinforced alumina ceramics combine crack bridging, fiber pullout, and crack deflection processes. Besides enhancement of mechanical performance, the addition of SiC whiskers also improves wear resistance compared with pure alumina, particularly in the severe wear range, due to the inhibition of crack propagation by the whiskers [3].

In our previous work [4], graphene-reinforced Al_2_O_3_-SiCw ceramic nanocomposites and the influence of the volume fraction of graphene oxide loading on the resulting mechanical and electrical properties were studied. It was found that the ceramic matrix with homogeneous dispersion of a small amount (0.5 vol. %) of graphene is more prone to produce various extrinsic reinforcement mechanisms (crack deflection, pull-out, and bridging), which are promoted due to the presence of graphene at the Al_2_O_3_-SiCw boundaries. According to the mechanical characterization, Al_2_O_3_-SiCw/Graphene nanocomposites have the highest flexural strength (904 MPa), fracture toughness (10.6 MPa∙m^1/2^), and hardness (22 GPa) in comparison with Al_2_O_3_/SiCw composites without graphene (750 MPa, 9.5 MPa∙m^1/2^ and 21.1 GPa, respectively).

In addition, graphene has a friction-reducing potential for solid lubricating and can decrease the friction force acting on the contact surfaces at the micro- and nano-scale [5,6]. 

Recent studies report on the enhancement of wear performance of ceramic-based nanocomposites in case graphene is added. Kim et al. studied wear and mechanical behavior of pressureless sintered Al_2_O_3_/Graphene nanocomposites. They found that the exfoliated graphene platelets were effective in enhancing both mechanical and wear resistance properties due to the lack of defects [7]. For spark plasma sintered SiC/Graphene nanocomposites, Llorente et al. reported an enhanced wear resistance SiC ceramics under dry sliding conditions due to the ability to form a wear protecting graphene-based tribolayer. However, the friction performance depended on the different kinds and amounts of graphene and sliding distance as well [8]. Rutkowski et al. observed that the friction coefficient decreased from 0.8 to 0.5 if 0.5 wt. % graphene was added to the Si_3_N_4_ matrix. A graphene content of more than 2 wt. % increases the material wear rates, which can be a result of the graphene removing process and filling the voids by the wearing off of fine silicon nitride particles on the track after the friction test [9]. Gutierrez-Gonzalez et al. found out that under the same testing conditions the alumina/graphene nanocomposites with an addition of 0.22 wt.% showed approximately half of the wear rate, and a 10% decrease of friction coefficient compared to the monolithic alumina. Decreasing in wear rate and friction coefficient was explained in terms of the presence of graphene platelets that act as a self-lubricating layer on the rubbing surface in comparison to the Al_2_O_3_/Al_2_O_3_ tribological system [10].

The aim of the current study is to investigate the wear behavior of Al_2_O_3_-SiCw nanocomposite with 0.5 vol. % graphene addition using dry ball-on-disk wear test under two (10 N and 40 N) sliding loads. The light load can be used to diagnose the onset of wear in the nano/micro-device industry, while higher load is important to study the friction and wear behavior of these nanocomposites for heavy applications (e.g., high temperature valve system, bearing, conveyor system, valves, etc.) and protective coating applications (e.g., armor, prostheses, etc.). 

## 2. Materials and Methods

### 2.1. Nanocomposite Preparation

The as-produced graphene oxide (GO) dispersion contained 1 mg/ml of mostly monolayer GO flakes with thickness ≤ 2 graphene layers. The commercially available Ceramtuff blend (grade HA9S), produced by the company Advanced Nanocomposite Materials, LLC (Greer, SC, USA) was mixed with 0.5% vol. amount of GO using a colloidal method. Disks of 20 mm diameter and 4 mm thickness were fabricated by spark plasma sintering (SPS, FCT Systeme GmbH, KCE FCT-H HP D-25 SD, Rauenstein, Germany) in vacuum at 1780 °C applying a heating rate of 100 °C/min and pressure of 80 MPa. It is important to note, that in our previous work it was found that the thermal reduction (including the restoration of large sp^2^ regions) of graphene oxide was favored by SPS at 1780 °C [4]. For comparison purposes, Al_2_O_3_-SiCw powders without GO were SPSed following the same sintering cycle. Details of the powder mixtures processing and sintering parameters were reported elsewhere [4].

### 2.2. XRD Characterization

XRD measurements were carried out in an Empyrean diffractometer (PANalytical, Almelo, Netherlands) ranging from 20° to 90° on the polished composite surfaces. The step size was 0.05° with a scan speed of 0.06°/min. The diffractometer used Cu Kα radiation (ʎ = 1.5405981), working at 60 kV and with an intensity of 30 mA.

### 2.3. Wear Test Set-Up and Conditions

The wear resistance of studied materials was evaluated under room-temperature dry conditions. A “ball-on-disk” type wear tests were carried out in a TETRA BASALT-N2 tribometer (TETRA GmbH, Ilmenau, Germany) in conformity with American Society for Testing and Materials (ASTM G99) requirements. The sintered samples were grinded and polished down to 1 μm prior to wear testing. All wear experiments were carried out against a pure alumina ball with a 3 mm diameter with a rotating speed of 3 rps and a radius of 5 mm.

The test duration was associated with a traveling distance (*S*) of 10 km. The applied load (*F_N_*) was 10 N and 40 N corresponding to initial Hertzian contact pressures of 2.1 GPa and 3.4 GPa, respectively. At least four wear tests were conducted from each composition. Prior to each wear test, the specimens were rinsed ultrasonically in ethanol for 15 min and air-dried using compressed air.

The wear rate (*W*) has been computed using Equation (1):(1)$$W=\frac{\Delta V}{F_{N}\cdot S}$$

Being ∆*V* the volume loss after the tests (mm^3^), *F_N_* the applied load (N), and *S* the sliding distance (m). The volume losses were accurately measured via a 3D surface profile of wear tracks, which was obtained using a profilometer Talysurf CLI 500 (Taylor Hobson, Leicester, UK) that maps the measured area by tip-sample surface contact, being the step 0.01 µm and the scanning speed 0.1 mm/s. The average surface roughness (Ra) of the worn surface after the wear test was analyzed at five different locations of the 3D wear track topographies. After each sliding test, the worn surfaces were blown out with the compressed air prior to scanning electron microscope VEGA 3 LMH (SEM Tescan, Brno, Czech Republic) observation. The images were obtained by means of a secondary electron detector under high vacuum mode (5 × 10^−2^ Pa) using accelerating voltage 5.0 kV, probe current 10 or 12 µA, with different magnifications.

The wear scars and polished surfaces were examined under a DXR^TM^2 (Thermo Fisher Scientific, MA, USA) confocal Raman microscope with an excitation wavelength of 532 nm and a laser power of 8.0 mW. The parameters of Raman configuration were stated in a previous work [4]. In addition, Raman spectra of the as-worn surface of counterparts were collected to confirm transfer of graphene or graphitic phases from the nanocomposites to the alumina ball. In this case, the laser beam was focused through an optical microscope’s 50× objective lens to a spot size of 50 µm on the studied area (from different spots, at an interval of 200 nm). The accumulation time for each Raman spectrum was about 10 s.

## 3. Results and Discussion

The Raman spectrum of the as-received graphene oxide and X-ray diffraction pattern corresponding to the as-received Al_2_O_3_-SiCw powder are presented in Figure 1. For graphene oxide, typical D and G peaks were observed at ~1350 cm^−1^ and ~1605 cm^−1^, respectively (Figure 1A). The D band confirms the lattice distortion due to the disorder in the sp^2^ structure whereas G band indicates the graphitic nature.

XRD pattern of the ceramic raw powder shows small broadening of peaks (Figure 1B), while after sintering peaks are narrower due to crystallization process (Figure 2). 

In addition, the pattern reveals that no contamination and/or presence of side reactions along the sintering of the powder was detected. The peaks corresponding to the graphene structure are not detectable by XRD due to the low content of graphene.

The thermally etched, polished sections of dense Al_2_O_3_-SiCw (A) and Al_2_O_3_-SiCw/Graphene (B) sintered composites where the SiCw grains are the elongated inclusions, which were studied by SEM (Figure 3). 

The alumina matrix in both materials exhibits a relatively broad grain size distribution but the mean particle size was similar (around 2 µm).

Figure 4A shows the friction coefficient as a function of the sliding distance registered during the wear test for the Al_2_O_3_-SiCw ceramic with and without graphene under dry conditions when worn against the alumina ball.

Under a load of 10 N, the friction coefficient for ceramic composites without and with graphene remains stable around 0.31 and 0.25, respectively. When the load was increased to 40 N the friction coefficient for Al_2_O_3_-SiCw ceramic increased up to 0.65 during the first meters of the experiment, and after this initial stage, the friction coefficient fixed on a steady level (≈0.58). This behavior can be attributed to a polishing process during the wear test where the surface asperities or roughness peaks are removed [11,12,13,14]. Meanwhile, in the case of the nanocomposite with graphene, the friction coefficient remains stable at 0.41. For ceramic/graphene nanocomposites, the low coefficient of friction can be attributed to the presence of the graphene flakes and the role that they play in the tribological system. It has been published elsewhere that graphene plays an important role in the rubbing behavior due to its lubricant nature that may reduce friction in the sliding interfaces [15,16]. Raman spectroscopy studies confirmed the presence of graphene or graphitic phases transferred from the nanocomposite to the worn surface of the alumina ball, and, consequently, the formation of the lubricating layer (Figure 4B). Figure 5 and Figure 6 show the surface topography of the three-dimensional wear tracks and the wear rates for the ceramic and ceramic/graphene materials studied after sliding under 10 N and 40 N load against pure alumina ball, respectively.

In both cases under identical conditions, the deepest wear track was measured for the Al_2_O_3_-SiCw composite. From the 3D wear track surface topographies, the wear volume loss (*W*) was estimated using Equation (1). The favorable effect of the presence of graphene in the ceramic matrix on the wear resistance of the nanocomposites is shown in Figure 5C and Figure 6C. The Al_2_O_3_-SiCw/Graphene nanocomposites under the two loads of 10 N and 40 N is almost double wear resistant compared to that of a matrix material. SEM analysis of the worn surfaces confirmed that the observed changes in wear rate and friction coefficient were due to a fundamental change in the process of wear. At low magnification, morphologies of wear track of both materials when applying the load of 10 N are smooth and look similar (Figure 7A,B). 

However, at high magnification the image of the wear track (Ra ~ 0.076 µm) of Al_2_O_3_-SiCw ceramic (Figure 8A) shows evidence of pull-out, while the wear scar of the Al_2_O_3_-SiCw/Graphene nanocomposite exhibits a smooth wear surface (Ra ~ 0.049 µm) as shown in Figure 8B.

On the other side, it is clear that the morphology of the wear tracks for both composites is different at a higher applied load (40 N) from a lower applied load (10 N), even at low magnification (Figure 7C,D). Additionally, it should be noted that obtained was a set of images that were similar and, therefore, we considered them as sufficiently representative.

Under high load, the texture of the wear surface consists of the repeated strips perpendicular to the sliding direction was observed. In addition, in a material with graphene, the friction surface presents periodic dark (2–5 μm width) and light (10–30 μm width) curve shape strips perpendicular to the sliding direction likely radius of the counterbody (Figure 7D).

As shown in Figure 8C, the composite without graphene worn surface generated under the high load was generally rough (Ra ~ 0.142 µm). Under these conditions, the pulled-out grains were observed, and the dominant material removal mechanism was determined as an intragranular fracture (Figure 8C). Meanwhile, for the nanocomposite with graphene under the same sliding conditions, a relatively smooth surface (Ra ~ 0.087 µm) was observed (Figure 8D).

The addition of graphene to the ceramic matrix significantly improve the wear resistance of the nanocomposites under high load due to less alumina and SiCw grain pull-out during the wear test. The reason for this behavior might be due to the presence of an auto-lubricating layer of graphene in the ceramic matrix that provides sufficient lubrication between the specimen and the alumina ball that acts as a counterpart material. Moreover, a higher fracture toughness and less tangential force applied to the alumina grains in nanocomposites with graphene contribute to the decrease in crack propagation during the test in comparison to the composites without graphene [4].

In order to study the role played by graphene in the wear process, wear tracks and debris were studied by micro-Raman spectroscopy (Figure 9).

Besides peaks like alumina at ~ 420 cm^−1^, silicon carbide at ~ 799 cm^−1^, the three typical peaks corresponding to the graphene-based structure at ∼ 1350 cm^−1^ (D band), ∼ 1585 cm^−1^ (G band) and ∼ 2700 cm^−1^ (2D band) were detected.

Figure 9 (left column) shows the comparison of the Raman spectra of the untested and worn surface of the composites without and with graphene addition. The intensity of the Al_2_O_3_ and SiC peaks inside and outside of the wear track of Al_2_O_3_-SiCw composites is similar, while inside the wear track of Al_2_O_3_-SiCw/Graphene nanocomposites, the scans revealed an increase in the D peak intensity. Thus, the increased D peak intensity inside the wear track of the Al_2_O_3_-SiCw/Graphene nanocomposite confirmed the higher percentage of the graphene surface exposed to the Raman measurement in the damaged surface and tribolayer formation during tribology testing on the Al_2_O_3_-SiCw/Graphene nanocomposites resulting in improved wear properties.

By merging both spectra, in particular, the SiC and graphene-based structures bands, and SEM photos of the analyzed area of the Al_2_O_3_-SiCw (top) and Al_2_O_3_-SiCw/Graphene (bottom) composites colored debris maps were created (Figure 9 right column). Green and yellow colored zones of the Raman maps of the composites show the presence of the D- and G-peaks and SiC peaks, respectively. As can be seen the whole damaged surface of the nanocomposite with graphene is green only with the small presence of SiC, which can provide experimental evidence of the graphene enriched layer caused by the sliding force and consequently self-organization phenomenon. Additionally, the periodic dark and light curve shape strips observed on the friction surface (Figure 7D) might be viewed as a trace of a self-organization response to the wear process; our group has discussed this view in detail elsewhere [17,18,19].

In summary, the present work demonstrates that the Al_2_O_3_-SiCw/Graphene nanocomposites show a lower friction coefficient and a higher wear resistance, related to the presence of an adhered tribolayer of graphene platelets on the worn surface of the alumina ball and composite that enhances the tribological performance of ceramic/graphene composites. Evidence for the formation of such tribofilm includes Raman spectroscopy mapping and SEM (Figure 9).

## 4. Conclusions

According to the results of the ball-on-disk wear test under dry conditions, Al_2_O_3_-SiCw/Graphene nanocomposites fabricated by Spark Plasma Sintering exhibited a lower wear rate and lower friction coefficient than the Al_2_O_3_-SiCw ceramic. Compared to the ceramic composites without graphene, for 0.5 vol. % graphene nanocomposites, the percentage decrease of friction coefficient and the wear rate were reported as 20% and 45% under low load (10 N), and 30% and 55% under high load (40 N), respectively. This improvement on the wear behavior is related to the presence of an auto-lubricating layer of graphene in the ceramic matrix that provides sufficient lubrication between the specimen and the alumina ball that acts as a counterpart material. The formation of such tribofilm is related to the presence of graphene platelets adhered to the friction surface, and was confirmed by Raman and SEM observations. 

## Figures and Tables

**Figure 1 nanomaterials-09-00151-f001:**
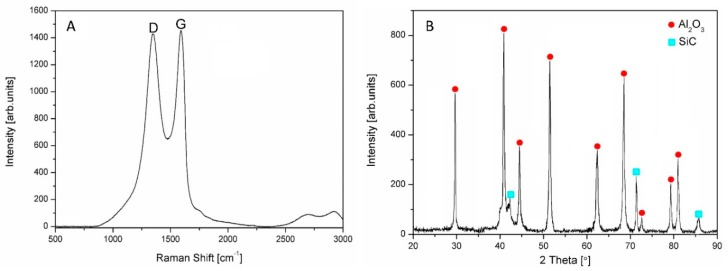
Raman spectrum of the as-received graphene oxide (**A**) and XRD pattern of the as-received Al_2_O_3_-SiCw powder (**B**).

**Figure 2 nanomaterials-09-00151-f002:**
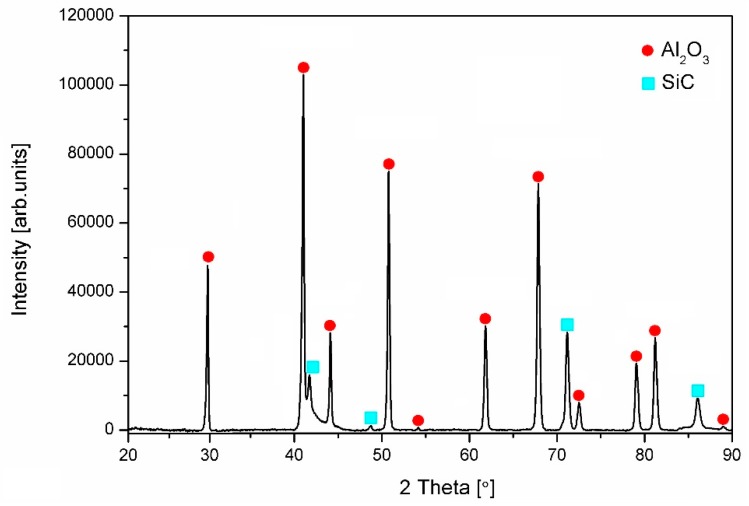
XRD pattern of the sintered and polished surface of nanocomposites.

**Figure 3 nanomaterials-09-00151-f003:**
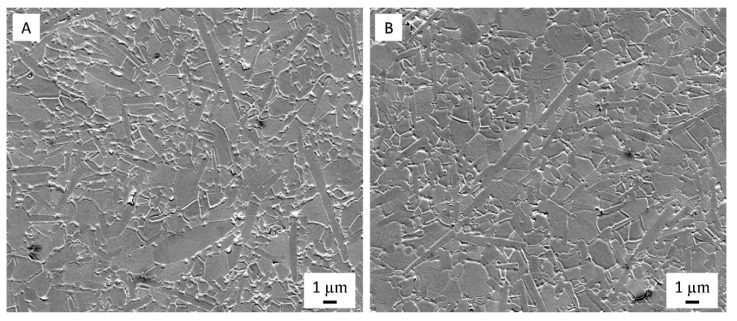
SEM images of polished and thermally etched sections for Al_2_O_3_-SiCw (**A**) and Al_2_O_3_-SiCw/Graphene (**B**) sintered composites.

**Figure 4 nanomaterials-09-00151-f004:**
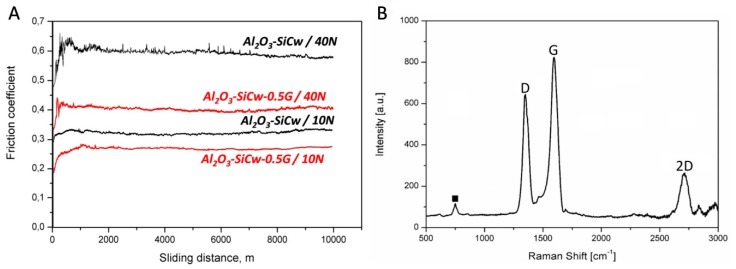
Friction coefficient as a function of the sliding distance registered during the wear test for both materials when worn against the alumina ball (**A**). Raman spectrum for worn surface of counterpart (Al_2_O_3_ ball), which shows transferred graphene (“D”, “G”, and “2D”) from the nanocomposites (**B**). “■” label denotes the alumina peak.

**Figure 5 nanomaterials-09-00151-f005:**
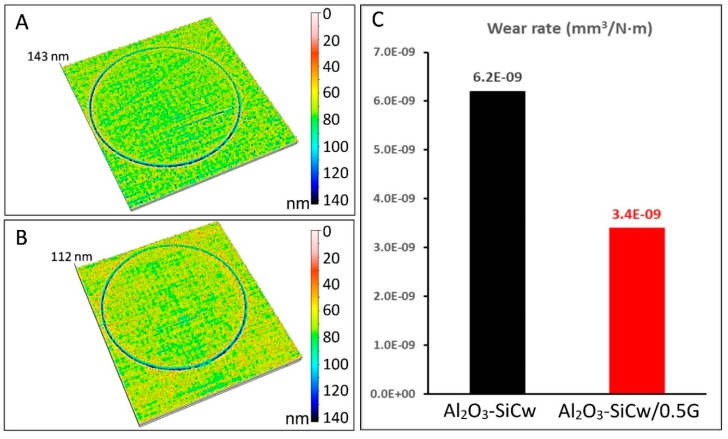
3D wear track topographies corresponding to (**A**) Al_2_O_3_-SiCw and (**B**) Al_2_O_3_-SiCw/Graphene composite as a function of depth (colored scale). Wear rate values under 10 N load of both materials after10 km of wear test (**C**).

**Figure 6 nanomaterials-09-00151-f006:**
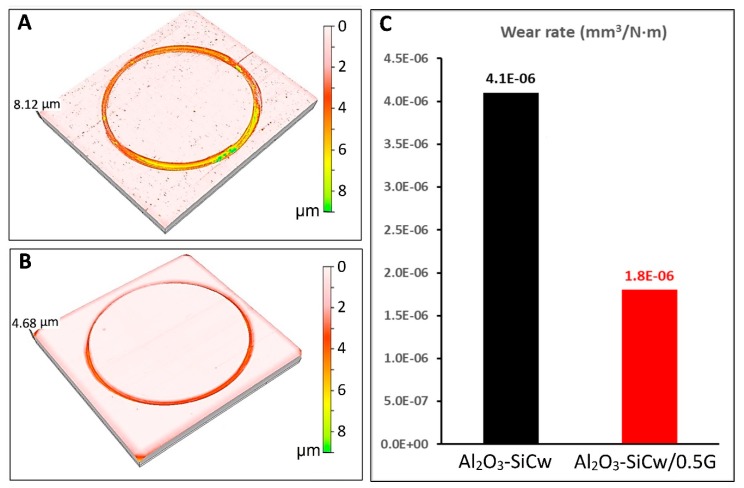
3D wear track topographies corresponding to (**A**) Al_2_O_3_-SiCw and (**B**) Al_2_O_3_-SiCw/Graphene composite as a function of depth (colored scale). Wear rate values under 40 N load of both materials after 10 km of wear test (**C**).

**Figure 7 nanomaterials-09-00151-f007:**
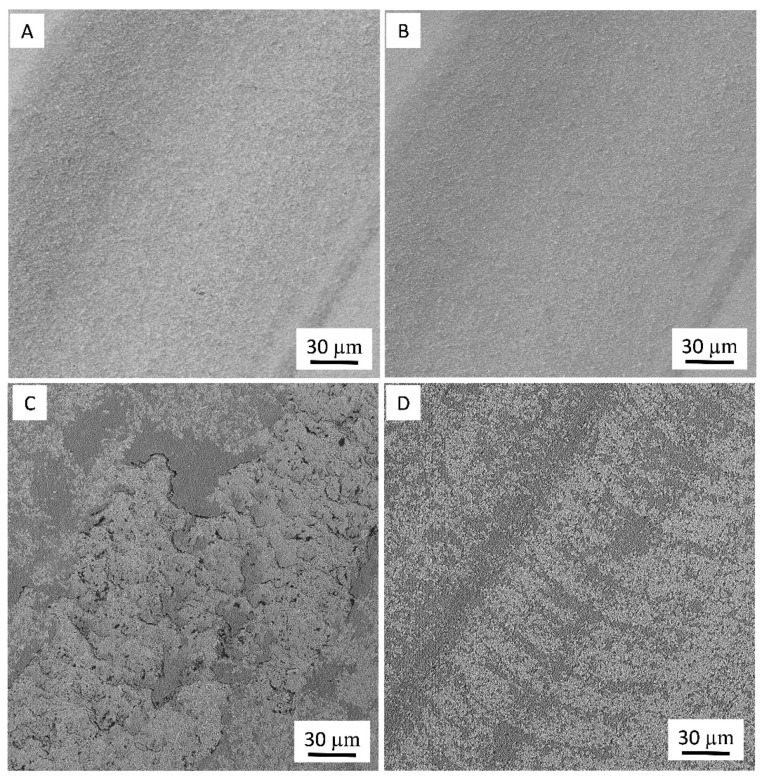
SEM micrographs of the worn surface of (**A**,**C**) Al_2_O_3_-SiCw and (**B**,**D**) Al_2_O_3_-SiCw/Graphene composites under load of 10 N and 40 N, respectively.

**Figure 8 nanomaterials-09-00151-f008:**
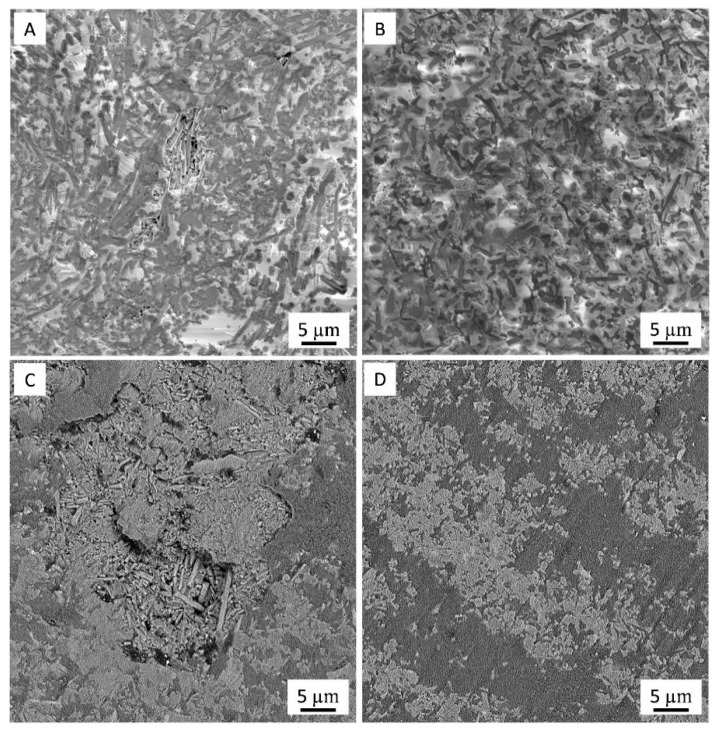
SEM close-up images of the worn surface of (**A**,**C**) Al_2_O_3_-SiCw and (**B**,**D**) Al_2_O_3_-SiCw/Graphene composites under load of 10 N and 40 N, respectively.

**Figure 9 nanomaterials-09-00151-f009:**
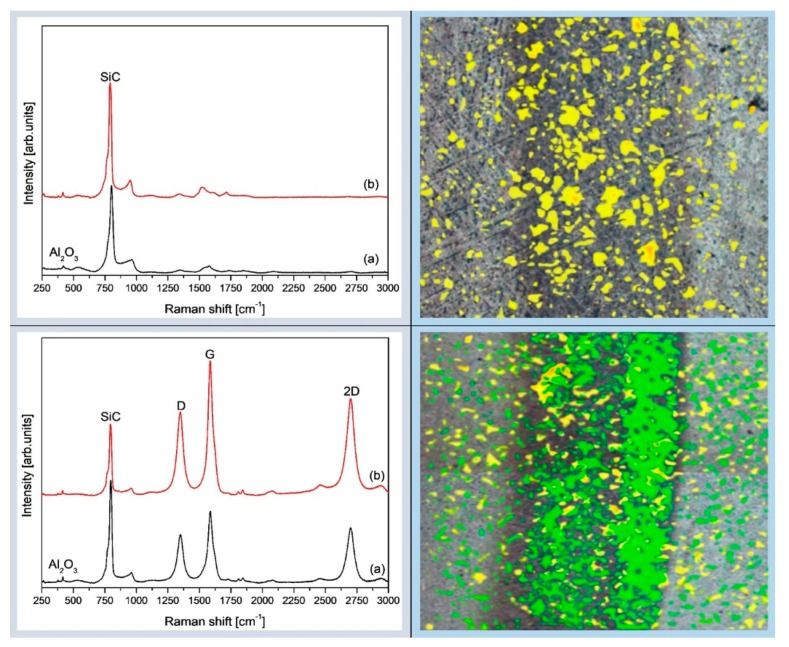
(**left column**) The comparison of the Raman scans of Al_2_O_3_-SiCw (**top**) and Al_2_O_3_-SiCw/Graphene (**bottom**) composites obtained from surfaces without (a) and with wear (b). (**right column**) Raman maps of the debris created by merging the intensity maps of the SiC band and graphene-based structures band for the original polished and wear track surfaces and SEM photos of the analyzed area of the Al_2_O_3_-SiCw (**top**) and Al_2_O_3_-SiCw/Graphene (**bottom**) composites. Green and yellow color features are related to D- and G-peaks, and SiC peaks, respectively.

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
