# Peer review of "Wear Behavior of Graphene-Reinforced Alumina–Silicon Carbide Whisker Nanocomposite"

_nanomaterials, 2019, doi:10.3390/nano9020151_

Round 1

Reviewer 1 Report

This is essentially a continuation of previous work by the same group [4 ] as regards the mechanical properties of  Al2O3/SiCw/grapheme nanocomposites. Here the authors focus on wear and the probing and interpretation of the mechanism responsible for the observed improvement (‘doubling’) of the wear resistance as a result of the 0.5 % grapheme addtion. The topic is attractive and the experimental work is well performed and well reported. The lubricating action of grapheme (which, after all, is a graphite layer) is reasonable.

Everythingn is fine up to the two thrids of page 13.Then the authors condsider their findings in terms of self-organization in a nearly pointless (certainly non-illuminating) fashion . Here, I understand that the authors do not suggest that  graphene is already [self-]organized before testing  but it is  mechanically forced to organize during friction testing.

Now most of the provided references pertaining to tribology + self-organization   are other works by the present authors and 3 of Prigogine’s general works on self-organization / dissipative structures etc.  I do not see how references [19], [20], [23]  (all of them in my library for several years and I am quite familiar with them) tell us something essential about the experimental findings in the work in consideration; for example, can we say on the basis of this background  something about the spacing strip pattern (e.g. can I say something about the absolute value of spacing ? how can I change the pattern in a desirable direction (and which  is that?)? Does equation (4) provide answers to my questions? [ I do not think so!]

Alternatively one can say that, more or less, exerted mechanical forces (shear etc) during testing cause the graphene flakes to arrange flat etc and thus friction is reduced; a simple interpretation of that kind can be polished somewhat and become accepted.

Overall, the paper can be accepted IF the authors are willing to prepare a shorter new version where the bulk of the self-organization considerations (last 4 pages before conclusions) is removed and replaced by a  half a page long text  (plus figure 9 if needed, though the two materials are indistinguishable in this figure (if error bars are added) and hence no one can conclude something as regards wear improvement upon grapheme addition from figure 9). In the revised manuscript reference to self-organization can be limited to statement such as the following one:

 The strip pattern observed in Fig, 7d might be viewed as a trace of a self-organization response to wear processes; our group has discussed this view in detail elsewhere ([16], [18], [22]).

I suggest this drastic condensation  because all I see in the aforementioned 4 pages is an unnecessary and obscure discussion undermining instead of enhancing a valuable piece of well performed and well described experimental work.

Additional points: (1) The second reference [21] should be reference [22]

(2) Fig. 7 d does not suffice for the claim that there is s strip pattern; a photo at lower magnification should be provided.

(3) Equation (6) is mistyped.

Author Response

Point 1: This is essentially a continuation of previous work by the same group [4] as regards the mechanical properties of Al2O3/SiCw/grapheme nanocomposites. Here the authors focus on wear and the probing and interpretation of the mechanism responsible for the observed improvement (‘doubling’) of the wear resistance as a result of the 0.5 % grapheme addition. The topic is attractive and the experimental work is well performed and well reported. The lubricating action of grapheme (which, after all, is a graphite layer) is reasonable.

Response 1: We thank the reviewer for carefully reading and summarizing the manuscript. We are glad to see that the reviewer has a positive opinion about our work.

Point 2: Everything is fine up to the two thirds of page 13. Then the authors consider their findings in terms of self-organization in a nearly pointless (certainly non-illuminating) fashion. Here, I understand that the authors do not suggest that   graphene is already [self-organized before testing but it is mechanically forced to organize during friction testing.

Now most of the provided references pertaining to tribology + self-organization are other works by the present authors and 3 of Prigogine’s general works on self-organization / dissipative structures etc. I do not see how references [19], [20], [23] (all of them in my library for several years and I am quite familiar with them) tell us something essential about the experimental findings in the work in consideration; for example, can we say on the basis of this background something about the spacing strip pattern (e.g. can I say something about the absolute value of spacing ? how can I change the pattern in a desirable direction (and which is that?)? Does equation (4) provide answers to my questions? [I do not think so!]. Alternatively one can say that, more or less, exerted mechanical forces (shear etc) during testing cause the graphene flakes to arrange flat etc and thus friction is reduced; a simple interpretation of that kind can be polished somewhat and become accepted.

Overall, the paper can be accepted IF the authors are willing to prepare a shorter new version where the bulk of the self-organization considerations (last 4 pages before conclusions) is removed and replaced by a half a page long text (plus figure 9 if needed, though the two materials are indistinguishable in this figure (if error bars are added) and hence no one can conclude something as regards wear improvement upon grapheme addition from figure 9). In the revised manuscript reference to self-organization can be limited to statement such as the following one:

The strip pattern observed in Fig, 7d might be viewed as a trace of a self-organization response to wear processes; our group has discussed this view in detail elsewhere ([16], [18], [22]).

I suggest this drastic condensation because all I see in the aforementioned 4 pages is an unnecessary and obscure discussion undermining instead of enhancing a valuable piece of well performed and well described experimental work.

Response 2: We considered the comments of the reviewer to reduce the length of our paper. In the revised manuscript we were able to condense the bulk of the self-organization considerations (last 4 pages before conclusions) to one paragraph as per the reviewer's suggestion (Pages 13-17). The part about thermal conductivity measurements was deleted as well (Page 5 and Fig.9 of original submission).

Page 5: Thermal conductivity measurements were deleted.

2.3. Thermal conductivity measurements

The heat measurements were carried out on Netzch LFA 467 HT apparatus (NETZSCH GmbH & Co, Selb, Germany) by laser flash method. The Pyroceram 9606 with a known coefficient of thermal expansion and specific heat has been used as reference material for determining the specific heat.

The thermal diffusivity (α) measurements for the reference and test material were carried out in an argon atmosphere from room temperature to 1000°C. For better absorption of the laser pulse energy the surface of the samples was coated with a thin graphite layer. For these measurements, samples were machined to 12.7 mm in diameter and 2.5 mm thick. At least three measurements at each temperature were recorded. The thermal conductivity (λ) was calculated for each sample from the following equation:

                          (2)

where α(T) thermal diffusivity (mm2/s), Cp (T) specific heat (J/g∙K), ρ(T) density of the material (g/cm3).

Pages 13-17: The bulk of the self-organization considerations (last 4 pages before conclusions) is removed.

“It is known that any tribosystem joints characteristics of an engineering human and natural (friction itself) systems [16,17]. The presence of an external stimulus is sufficient for the response of the material, i.e., to change its properties. This responsiveness is an effective method to enhance the tribological properties of a material, and, consequently, reduce friction coefficient and improve wear resistance. This structural adaptability is closely linked with self-organization phenomenon which can lead to the formation of new dissipative structures (protective tribofilms) at the friction interface [18-21]. Therefore, the friction coefficient and wear rate can be further reduced by the formation of a lubricating graphene layer on the surface of the rubbing parts. In addition, friction also leading to a distribution of graphene in a rubbing body. Graphene spontaneously concentrates on the friction surface and its content is substantially higher than on the pristine surface. This distribution causes loss of entropy of rubbing body. Consequently, the graphene transfer process is accompanied by a negative production of entropy. The presence of such process can be a sign of self-organization phenomenon and leads to a lower material wear rate [18,22]. The self-organization process during friction was considered from the nonequilibrium thermodynamics and self-organization theory of Prigogine [23]. Based on this theory and our previous work [24] the probability of the stability loss in the tribosystem from the non-equilibrium thermodynamics and self-organization theory point of view was evaluated.

However, unlike previous study [24] when only one independent friction process is in progress in the tribosystem was reviewed, here we assume that two main processes (friction and mass-transfer to friction surface) take place in the tribosystem, consequently, entropy production will be expressed as follows:

                   (3)

where Jh—heat flux, Xh—thermodynamic force inducing heat flux equaling to gradT/T2, Jh =-λFgradT, along with this, Jh = kpv (k—friction factor, p—load, v—sliding velocity), T—temperature, λ—coefficient of thermal conductivity, X – thermodynamic force of mass-transfer, ρ – density of material transferred to friction surface, w – velocity of mass-transfer.

In accordance with the Equations (2) – (5) presented in our previous work [24], assume that sliding velocity and temperature do not depend on the load. Therefore, excess entropy production would appear as follows:

 (4)

Figure 2 shows that friction coefficient increases with applied load, therefore derivative k/p > 0. Consequently, the first term of the right part of the equation (4) could be negative if:

                                (5)

To fulfill the Equation (4), the coefficient of thermal conductivity should increase with the load. Thermodynamic force, causing mass transfer, as a rule, will increase with increasing load. The growth of the load will lead to an increase in mechanical stresses that cause deformation in the surface and near-surface layers of a rubbing body. Also, load increases will result in a higher temperature gradient and, correspondingly, the gradient of the chemical potentials that cause diffusion. Consequently, the derivative x/p > 0. The same reasoning can be given about the transfer rate, i.e. the growth of the thermodynamic force that causes mass transfer leads to an increase in the mass transfer rate, i.e. derivative w/p > 0. The area of contact increases with increasing load, i.e. derivative F/p > 0. Consequently, the second term on the right-hand side of equation (4) can become negative under the condition:

                                      (6)

Thus, self-organization can occur in a given tribosystem, subject to the conditions equations (5) and (6).

The thermal diffusivity was measured for each composition in the temperature interval between 25 °C and 1000 °C in the atmosphere of argon and thermal conductivity of the Al2O3/SiCw composites with and without graphene was calculated from Eq. (2) and presented in Figure 9.

Figure 9. The thermal conductivity of the Al2O3-SiCw composites with and without graphene oxide addition.

The measured data of thermal conductivity at room temperature were found in good agreement with literature experimental values for Al2O3-based composites with various volume fractions of SiC inclusions or whiskers (32–40 W/m∙K) [25,26]. Meanwhile, it was found that the presence of low graphene contents in Al2O3-SiCw ceramic matrix improves the thermal conductivity of nanocomposite compared to ceramic composites without graphene.

The increase in the thermal conductivity equation (5) of surface structures (secondary structures) can occur when they are enriched with graphene, which has an anomalously high thermal conductivity [27]. Reduction in the density of the transportable substances equation (6) can occur with the preferential transfer of graphene, which is the lightest phase component of the material of a rubbing body.

Thus, the reinforcement of the ceramic matrix with graphene is able to promote self-organization during friction. Friction-induced self-organization at the frictional interface of the graphene containing material allows wear reduction about two times compared to the material without graphene. The most likely mechanism is as follows. When the ball slides in front of it, a "roller" is formed from the material Al2O3-SiCw or material Al2O3-SiCw/Graphene. After a certain period of time, the "roller" or part of it breaks away from the body. Thus, the material wears out and the process repeats. The line of separation in this case ideally lies perpendicular to the direction of sliding of the ball and it occurs along the particle boundaries: Therefore, on the material Al2O3-SiCw, it has only a tendency to the ball breaks off the “roller” (Fig. 6C). Such a wear mechanism is characteristic of the material Al2O3-SiCw. On the other hand, in case of Al2O3-SiCw/Graphene material, during sliding of alumina ball, there is a local decrease in the coefficient of friction, and then the ball can jump over the roller. The friction coefficient decreases (Fig. 3A) due to the presence of graphene at the boundary of the particle and a banded structure is formed with a curved morphology perpendicular to the sliding direction (Fig. 6D, Fig. 7D). In regions where there is not enough graphene on the surface, the detachment of the particles occurs according to the mechanism characteristic of the Al2O3-SiCw material. This mechanism of wear and the predominant mass transfer of graphene to the friction surface are characteristic for the Al2O3-SiCw/Graphene material.

Furthermore, Raman spectra show the presence of graphene or graphitic phases on the worn surface of an alumina ball (Fig. 3B). This also endorses the idea that sliding wear mechanism dominates in graphene containing ceramic rather than abrasion or pull-out, and, consequently, leads to improvement in wear resistance of Al2O3-SiCw/Graphene nanocomposite.”

Page 13 lines 21-26 and Page 14 lines 1-2: The new text was added

Additionally, the periodic dark and light curve shape strips observed on the friction surface (Fig. 7D) might be viewed as a trace of a self-organization response to wear process; our group has discussed this view in detail elsewhere [17-19].

In summary, the present work demonstrates that Al2O3-SiCw/Graphene nanocomposites show a lower friction coefficient and higher wear resistance, related to the presence of an adhered tribolayer of graphene platelets on the worn surface of the alumina ball and composite that enhances tribological performance of ceramic/graphene composites. Evidence for the formation of such tribofilm includes Raman spectroscopy mapping and SEM (Fig.9).”

Point 3: The second reference [21] should be reference [22].

Response 3: The reference [21] was renumbered to reference [19] due to shortened of a manuscript.

Point 4: Fig. 7 d does not suffice for the claim that there is s strip pattern; a photo at lower magnification should be provided.

Response 4: The text was changed to show the photo at lower magnification.

Page 13 line 22

“…surface (Fig. 7D) might be viewed…

Point 5: (3) Equation (6) is mistyped.

Response 5: Equation (6) was deleted due to shortened of a manuscript.

Reviewer 2 Report

The manuscript on the friction study of ceramic composite enriched with graphene particles is well designed experimental work with the history of the research on this material. The articles topic is important for the application sphere  and research in friction materials.

Very well written introduction brings the reader to the key moments of the work.

However, the methods are missing description of the analytical tools: XRD tool and conditions of measurement and SEM conditions of image acquiring condition.  In results composite pattern range is should be from 20 2theta degree. Very important to understand the effect of sintration.

It would be more clear, if as received materials characterization (XRD, Raman) would be provided.

And there is issue of graphene, it is very difficult to understand the actual form of Graphene in the process. In the beginning it is graphene oxide, as I read from reference 4 , Hummers method prepared, the chemical reduction was not preformed here. Then in, ref.4 abs, authors state that  ..." plasma sintering method has the ability to reduce GO to graphene"  . I believe that some case of reduction of most functional groups created by drastic oxidation is removed, but nice graphene layer without oxide and holes wont be obtained (Bianco et al.Carbon 2013, 65, 1-6). 

It would be nice to study the debris of the friction process and using e.g. XPS to explain the actual state of C.

The impact of graphene layer sliding is evident just not explained the cause of the effect.

Author Response

Point 1: The manuscript on the friction study of ceramic composite enriched with graphene particles is well designed experimental work with the history of the research on this material. The articles topic is important for the application sphere and research in friction materials. Very well written introduction brings the reader to the key moments of the work.

Response 1: We thank the reviewer for his/her careful review and for positive comments.

Point 2: However, the methods are missing description of the analytical tools: XRD tool and conditions of measurement and SEM conditions of image acquiring condition. In results, composite pattern range is should be from 20 2theta degree. Very important to understand the effect of sintration. It would be more clear, if as received materials characterization (XRD, Raman) would be provided.

Response 2: XRD equipment and conditions of measurement and SEM conditions of image acquiring condition description was added. The characterization of as-received materials was provided.

Page 4 lines 9-13

2.2. XRD characterization

XRD measurements were carried out in an Empyrean diffractometer (PANalytical, Almelo, Netherlands) ranging from 20° to 90° on the polished composite surfaces. The step size was 0.05° with a scan speed of 0.06 °/min. The diffractometer used Cu Kα radiation (ʎ = 1.5405981), working at 60 kV and with an intensity of 30 mA.”

Page 5 lines 9-11

“The images were obtained by means of a secondary electron detector under high vacuum mode (5∙10-2 Pa) using accelerating voltage 5.0 kV, probe current 10 or 12 µA, with different magnifications.”

Page 5 lines 22-24 and Page 6 lines 1-8

The D band confirms the lattice distortion due to the disorder in the sp2 structure whereas G band indicates the graphitic nature. XRD pattern of the ceramic raw powder shows a small broadening of peaks (Fig. 1B), while after sintering peaks are more narrow due to crystallization process (Fig. 2).”

Point 3: And there is issue of graphene, it is very difficult to understand the actual form of Graphene in the process. In the beginning it is graphene oxide, as I read from reference 4, Hummers method prepared, the chemical reduction was not preformed here. Then in, ref.4 abs, authors state that ..." plasma sintering method has the ability to reduce GO to graphene". I believe that some case of reduction of most functional groups created by drastic oxidation is removed, but nice graphene layer without oxide and holes wont be obtained (Bianco et al.Carbon 2013, 65, 1-6). 

Response 3: We thank the reviewer for this observation. As reported in relevant literature high temperature reached in a short period of time during SPS technique has proven to be sufficient for an in situ reduction of graphene oxide which allows avoiding an extra reduction step [1-6]. The new text was added.

[1] Porwal, H.; Grasso, S.; Mani, M.K.; Reece, M.J. In situ reduction of graphene oxide nanoplatelet during spark plasma sintering of a silica matrix composite. J. Eur. Ceram. Soc. 2014, 34, 3357–3364; https://doi.org/10.1016/j.jeurceramsoc.2014.04.031

[2] Dong, L.L.; Xiao, B.; Liu, Y.; Li, Y.L.; Fu, Y.Q.; Zhao, Y.Q.; Zhang, Y.S. Sintering effect on microstructural evolution and mechanical properties of spark plasma sintered Ti matrix composites reinforced by reduced graphene oxides. Ceram. Int. 2018, 15,17835-17844; https://doi.org/10.1016/j.ceramint.2018.06.252

[3] Ramı́rez, C.; Vega-Diaz, S.F.; Morelos-Gómez, A.; Figueiredo, F.M.; Terrones, M.; Osendi, M.I.; Belmonte, M.; Miranzo, P. Synthesis of conducting graphene/Si3N4 composites by spark plasma sintering. Carbon 2013, 57, 425-432; https://doi.org/10.1016/j.carbon.2013.02.015

[4] Li, Z.; Tang, X-Z.; Zhu, W.; Thompson, B.C.; Huang, M.; Yang, J.; Hu, X.; Khor, K.A. Single-step process toward achieving superhydrophobic reduced graphene oxide. ACS Appl. Mater. Interfaces, 2016, 8, 10985–10994; DOI: 10.1021/acsami.6b01227

[5] Centeno, A.; Rocha, V.G.; Alonso, B.; Fernández, A.; Gutierrez-Gonzalez, C.F.; Torrecillas, R.; Zurutuza, A. Graphene for tough and electroconductive alumina ceramics. J. Eur. Ceram. Soc. 2013, 33; 3201–3210. https://doi.org/10.1016/j.jeurceramsoc.2013.07.007

[6] Miranzo, P.; Ramirez, C.; Roman-Manso, B.; Garzon, L.; Gutierrez, H.R.; Terrones, M.; Ocal, C.; Osendi, M.I.; Belmonte, M. In situ processing of electrically conducting graphene/SiC nanocomposites. J. Eur. Ceram. Soc. 2013, 33, 1665–1674; https://doi.org/10.1016/j.jeurceramsoc.2013.01.021

Page 4 lines 4-5

It is important to note, that in our previous work it was found that the thermal reduction (including the restoration of large sp2regions) of graphene oxide is favoured by SPS at 1780°C [4].

Point 4: It would be nice to study the debris of the friction process and using e.g. XPS to explain the actual state of C.

Response 4: We thank the reviewer for bringing up this point and we agree with him that an XPS analysis should be used for determining the actual state of C. However, at this moment we are unable to provide XPS study. Thus in the present manuscript, the study the debris of the friction process was carried out with SEM and Raman techniques (Pages 10-14 of the revised manuscript).

Point 5: The impact of graphene layer sliding is evident just not explained the cause of the effect.

Response 5: The following paragraph has been added and the cause of the graphene sliding effect was explained by the presence of an adhered tribolayer of graphene platelets on the worn surface of an alumina ball and composite

Page 13 lines 24-26 and Page 14 lines 1-2

In summary, the present work demonstrates that Al2O3-SiCw/Graphene nanocomposites show a lower friction coefficient and higher wear resistance, related to the presence of an adhered tribolayer of graphene platelets on the worn surface of the alumina ball and composite that enhances the tribological performance of ceramic/graphene composites. Evidence for the formation of such tribofilm includes Raman spectroscopy mapping and SEM (Fig.9).”

Round 2

Reviewer 1 Report

This is an interesting and well presented work with no pbjectionable and/or unnecessary parts; it can be published as it stands.